# Direct Optimization through $\arg\max$ for Discrete Variational Auto-Encoder

**Guy Lorberbom**
Technion

**Andreea Gane**
MIT

**Tommi Jaakkola**
MIT

**Tamir Hazan**
Technion

## Abstract

Reparameterization of variational auto-encoders with continuous random variables is an effective method for reducing the variance of their gradient estimates. In the discrete case, one can perform reparametrization using the Gumbel-Max trick, but the resulting objective relies on an $\arg\max$ operation and is non-differentiable. In contrast to previous works which resort to *softmax*-based relaxations, we propose to optimize it directly by applying the *direct loss minimization* approach. Our proposal extends naturally to structured discrete latent variable models when evaluating the $\arg\max$ operation is tractable. We demonstrate empirically the effectiveness of the direct loss minimization technique in variational autoencoders with both unstructured and structured discrete latent variables.

## 1 Introduction

Models with discrete latent variables drive extensive research in machine learning applications, including language classification and generation [42, 11, 34], molecular synthesis [19], or game solving [25]. Compared to their continuous counterparts, discrete latent variable models can decrease the computational complexity of inference calculations, for instance, by discarding alternatives in hard attention models [21], they can improve interpretability by illustrating which terms contributed to the solution [27, 42], and they can facilitate the encoding of inductive biases in the learning process, such as images consisting of a small number of objects [8] or tasks requiring intermediate alignments [25]. Finally, in some cases, discrete latent variables are natural choices, for instance when modeling datasets with discrete classes [32, 12, 23].

Performing maximum likelihood estimation of latent variable models is challenging due to the requirement to marginalize over the latent variables. Instead, one can maximize a variational lower-bound to the data log-likelihood, defined via an (approximate) posterior distribution over the latent variables, an approach followed by latent Dirichlet allocation [3], learning hidden Markov models [28] and variational auto-encoders [16]. The maximization can be carried out by alternatively computing the (approximate) posterior distribution corresponding to the current model parameters estimate, and estimating the new model parameters. Variational auto-encoders (VAEs) are generative latent variable models where the approximate posterior is a (neural network based) parameterized distribution which is estimated jointly with the model parameters. Maximization is performed via stochastic gradient ascent, provided that one can compute gradients with respect to both the model parameters and the approximate posterior parameters.

Learning VAEs with discrete $n$-dimensional latent variables is computationally challenging since the size of the support of the posterior distribution is exponential in $n$. Although the score function estimator (also known as REINFORCE) [39] enables computing the required gradients with respect to the approximate posterior, in both the continuous and discrete latent variable case, it is known to have high-variance. The reparametrization trick provides an appealing alternative to the score function estimator and recent work has shown its effectiveness for continuous latent spaces [17, 30]. In the discrete case, despite being able to perform reparametrization via the Gumbel-Max trick, the resulting

mapping remains non-differentiable due to the presence of $\arg\max$ operations. Recently, Maddison et al. [23] and Jang et al. [12] have used a relaxation of the reparametrized objective, replacing the $\arg\max$ operation with a *softmax* operation. The proposed *Gumbel-Softmax* reformulation results in a smooth objective function, similar to the continuous latent variable case. Unfortunately, the softmax operation introduces bias to the gradient computation and becomes computationally intractable when using high-dimensional structured latent spaces, because the softmax normalization relies on a summation over all possible latent assignments.

This paper proposes optimizing the reparameterized discrete VAE objective directly, by using the *direct loss minimization* approach [24, 14, 35], originally proposed for learning discriminative models. The cited work proves that a (biased) gradient estimator of the $\arg\max$ operation can be obtained from the difference between two maximization operations, over the original and over a perturbed objective, respectively. We apply the proposed estimator to the $\arg\max$ operation obtained from applying the Gumbel-Max trick. Compared to the Gumbel-Softmax estimator, our approach relies on maximization over the latent variable assignments, rather than summation, which is computationally more efficient. In particular, performing maximization exactly or approximately is possible in many structured cases, even when summation remains intractable. We demonstrate empirically the effectiveness of the direct optimization technique to high-dimensional discrete VAEs, with unstructured and structured discrete latent variables.

Our technical contributions can be summarized as follows: (1) We apply the direct loss minimization approach to learning generative models; (2) We provide an alternative proof for the direct loss minimization approach, which does not rely on regularity assumptions; (3) We extend the proposed direct optimization-based estimator to discrete VAEs with structured latent spaces.

## 2   Related work

Reparameterization is an effective method to reduce the variance of gradient estimates in learning latent variable models with continuous latent representations [17, 30, 29, 2, 26, 10]. The success of these works led to reparameterization approaches in discrete latent spaces. Rolfe et al. [32] and Vahdat and collaborators [38, 37, 1] represent the marginal distribution per binary latent variable with a continuous variable in the unit interval. This reparameterization approach allow propagating gradients through the continuous representation, but these works are restricted to binary random variables, and as a by-product, they require high-dimensional representations for which inference is exponential in the dimension size. Djolonga and Krause used the Lovasz extension to relax a discrete submodular decision in order to propagate gradients through its continuous representation [7].

Most relevant to our work, Maddison et al. [23] and Jang et al. [12] use the Gumbel-Max trick to reparameterize the discrete VAE objective, but, unlike our work, they relax the resulting formulation, replacing the $\arg\max$ with a softmax operation. In particular, they introduce the continuous Concrete (Gumbel-Softmax) distribution and replace the discrete random variables with continuous ones. Instead, our reparameterized objective remains non-differentiable and we use the direct optimization approach to propagate gradients through the $\arg\max$ using the difference of two maximization operations.

Recent work [25, 5] tackles the challenges associated with learning VAEs with structured discrete latent variables, but they can only handle specific structures. For instance, the Gumbel-Sinkhorn approach [25] extends the Gumbel-Softmax distribution to model permutations and matchings. The Perturb-and-Parse approach [5] focuses on latent dependency parses, and iteratively replaces any $\arg\max$ with a softmax operation in a spanning tree algorithm. In contrast, our framework is not restricted to a particular class of structures. Similar to our work, Johnson et al. [13] use the VAE encoder network to compute local potentials to be used in a structured potential function. Unlike the cited work, which makes use of message passing in graphical models with conjugacy structure, we use the Gumbel-Max trick, which enables us to apply our method whenever the two maximization operations can be computed efficiently.

## 3   Background

To model the data generating distribution, we consider samples $S = \{x_1, ..., x_m\}$ from a potentially high-dimensional set $x_i \in \mathcal{X}$, originating from an unknown underlying distribution. We estimate the

parameters $\theta$ of a model $p_\theta(x)$ by minimizing its negative log-likelihood. We consider latent variable models of the form $p_\theta(x) = \sum_{z \in \mathcal{Z}} p_\theta(z)p_\theta(x|z)$, with high-dimensional discrete variables $z \in \mathcal{Z}$, whose log-likelihood computation requires marginalizing over the latent representation. Variational autoencoders utilize an auxiliary distribution $q_\phi(z|x)$ to upper bound the negative log-likelihood of the observed data points:

$$\sum_{x \in S} -\log p_\theta(x) \leq \sum_{x \in S} -\mathbb{E}_{z \sim q_\phi} \log p_\theta(x|z) + \sum_{x \in S} KL(q_\phi(z|x)\|p_\theta(z)). \tag{1}$$

In discrete VAEs, the posterior distribution $q_\phi(z|x)$ and the data distribution conditioned on the latent representation $p_\theta(x|z)$ are modeled via the Gibbs distribution, namely $q_\phi(z|x) = e^{h_\phi(x,z)}$ and $p_\theta(x|z) = e^{f_\theta(x,z)}$. We use $h_\phi(x, z)$ and $f_\theta(x, z)$ to denote the (normalized) log-probabilities. Both quantities are modeled via differentiable (neural network based) mappings.

Parameter estimation of $\theta$ and $\phi$ is carried out by performing gradient descent on the right-hand side of Equation (1). Unfortunately, computing the gradient of the first term $\mathbb{E}_{z \sim q_\phi} \log p_\theta(x|z)$ in a high-dimensional discrete latent space $z = (z_1, ..., z_n)$ is challenging because the expectation enumerates over all possible latent assignments:

$$\nabla_\phi \mathbb{E}_{z \sim q_\phi} \log p_\theta(x|z) = \sum_{z \in \mathcal{Z}} e^{h_\phi(x,z)} \nabla_\phi h_\phi(x, z) f_\theta(x, z) \tag{2}$$

Alternatively, the score function estimator (REINFORCE) requires sampling from the high-dimensional structured latent space, which can be computationally challenging, and has high-variance, necessitating many samples.

## 3.1 Gumbel-Max reparameterization

The Gumbel-Max trick provides an alternative representation of the Gibbs distribution $q_\phi(z|x)$ that is based on the extreme value statistics of Gumbel-distributed random variables. Let $\gamma$ be a random function that associates an independent random variable $\gamma(z)$ for each input $z \in \mathcal{Z}$. When the random variables follow the zero mean Gumbel distribution law, whose probability density function is $g(\gamma) = \prod_{z \in \mathcal{Z}} e^{-(\gamma(z)+c+e^{-(\gamma(z)+c)})}$ for the Euler constant $c \approx 0.57$, we obtain the following identity[1] (cf. [18]):

$$e^{h_\phi(x,z)} = \mathbb{P}_{\gamma \sim g}[z^* = z], \text{ where } z^* \triangleq \arg\max_{\hat{z} \in \mathcal{Z}}\{h_\phi(x, \hat{z}) + \gamma(\hat{z})\} \tag{3}$$

Notably, samples from the Gibbs distribution can be obtained by drawing samples from the Gumbel distribution (which does not depend on learnable parameters) and applying a parameterized mapping, based on the $\arg\max$ operation. For completeness, a proof for the above equality appears in the supplementary material.

In the context of variational autoencoders, the Gumbel-Max formulation enables rewriting the expectation $\mathbb{E}_{z \sim q_\phi} \log p_\theta(x|z)$ with respect to the Gumbel distribution, similar to the application of the reparametrization trick in the continuous latent variable case [16]. Unfortunately, the parameterized mapping is non-differentiable, as the $\arg\max$ function is piecewise constant. In response, the Gumbel-Softmax estimator [23, 12] approximates the $\arg\max$ via the *softmax* operation

$$\mathbb{P}_{\gamma \sim g}[z^* = z] = \mathbb{E}_{\gamma \sim g}[\mathbf{1}_{z^* = z}] \quad \approx \quad \mathbb{E}_{\gamma \sim g} \frac{e^{(h_\phi(x,z)+\gamma(z))/\tau}}{\sum_{\hat{z} \in \mathcal{Z}} e^{(h_\phi(x,\hat{z})+\gamma(\hat{z}))/\tau}} \tag{4}$$

for a temperature parameter $\tau$ (treated as a hyper-parameter), which produces a smooth objective function. Nevertheless, the approximated Gumbel-Softmax objective introduces bias, uses continuous rather than discrete variables (requiring discretization at test time), and its dependence on the softmax function can be computationally prohibitive when considering structured latent spaces $z = (z_1, ..., z_n)$, as the normalization constant in Equation (4) sums over all the possible latent variable realizations $\hat{z}$.

## 3.2 Direct loss minimization

The direct loss minimization approach has been introduced for learning discriminative models [24, 14, 35]. In the discriminative setting, the goal is to estimate a set of parameters[2] $\phi$, used to predict a label for each (high-dimensional) input $x \in \mathcal{X}$ via $y^* = \arg\max_{y \in \mathcal{Y}} h_\phi(x, y)$, where $\mathcal{Y}$ is the set of continuous or discrete candidate labels. The score function $h_\phi(x, y)$ can be non-linear as a function of the parameters $\phi$, as developed by [14, 35].

Given training data tuples $(x, y)$ sampled from an unknown underlying data distribution $D$, the goodness of fit of the learned predictor is measured by a loss function $f(y, y^*)$, which is not necessarily differentiable. This is the case, for instance, when the labels $\mathcal{Y}$ are discrete, such as object labels in object recognition or action labels in action classification in videos [35]. As a result, the expected loss $\mathbb{E}_{(x,y)\sim D}[f(y, y^*)]$ cannot always be optimized using standard methods such as gradient descent.

The typical solution is to replace the desired objective with a surrogate differentiable loss, such as the cross-entropy loss between the targets and the predicted distribution over labels. However, the direct loss minimization approach proposes to minimize the desired objective directly. The proposed gradient estimator uses a loss-perturbed predictor $y^*(\epsilon) = \arg\max_{\hat{y}}\{h_\phi(x, \hat{y}) + \epsilon f(y, \hat{y})\}$ and takes the following form:

$$\nabla_\phi E_{(x,y)\sim D}[f(y, y^*)] = \lim_{\epsilon \to 0} \frac{1}{\epsilon}\Big( E_{(x,y)\sim D}[\nabla_\phi h_\phi(x, y^*(\epsilon)) - \nabla_\phi h_\phi(x, y^*)] \Big) \qquad (5)$$

In other words, the gradient estimator is obtained by performing pairs of maximization operations, one over the original objective (second term) and one over a perturbed objective (first term). The unbiased estimator is obtained when the perturbation parameter $\epsilon$ is approaching $0$. In practice, the parameter $\epsilon$ is assigned a small value, treated as a hyper-parameter, which introduces bias.

Unfortunately, the standard direct loss minimization approach predicts a single label $y^*$ for an input $x$ and, therefore, cannot generate a posterior distribution over samples $y$, i.e., it lacks a generative model. In our work we inject the Gumbel random variable to create a posterior over the label space enabling the application of this method to learning generative models. The Gumbel random variable allows us to overcome the general position assumption and the regularity conditions of [24, 14, 35].

## 4 Gumbel-Max reparameterization and direct optimization

We use the Gumbel-Max trick to rewrite the expected log-likelihood in the variational autoencoder objective $\mathbb{E}_{z\sim q_\phi} \log p_\theta(x|z)$ in the following form:

$$\mathbb{E}_{z\sim q_\phi} \log p_\theta(x|z) = \sum_{z\in\mathcal{Z}} \mathbb{P}_{\gamma\sim g}[z^* = z]f_\theta(x, z) = \mathbb{E}_{\gamma\sim g}[f_\theta(x, z^*)] \qquad (6)$$

where $z^*$ is the maximizing assignment defined in Equation (3). The equality results from the identity $\mathbb{P}_{\gamma\sim g}[z^* = z] = \mathbb{E}_{\gamma\sim g}[\mathbf{1}_{z^*=z}]$, the linearity of expectation $\sum_{z\in\mathcal{Z}} \mathbb{E}_{\gamma\sim g}[\mathbf{1}_{z^*=z}]f_\theta(x, z) = \mathbb{E}_{\gamma\sim g}[\sum_{z\in\mathcal{Z}} \mathbf{1}_{z^*=z}f_\theta(x, z^*)]$ and the fact that $\sum_{z\in\mathcal{Z}} \mathbf{1}_{z^*=z} = 1$.

The gradient of $f_\theta(x, z^*)$ with respect to the decoder parameters $\theta$ can be derived by the chain rule. The main challenge is evaluating the gradient of $\mathbb{E}_{\gamma\sim g}[f_\theta(x, z^*)]$ with respect to the encoder parameters $\phi$, since $z^*$ relies on an $\arg\max$ operation which is not differentiable. Our main result is presented in Theorem 1 and proposes a gradient estimator for the expectation $\mathbb{E}_{\gamma\sim g}[f_\theta(x, z^*)]$ with respect to the encoder parameters $\phi$. In the following, we omit $\gamma \sim g$ to avoid notational overhead.

**Theorem 1.** *Assume that $h_\phi(x, z)$ is a smooth function of $\phi$. Let $z^* \triangleq \arg\max_{\hat{z}\in\mathcal{Z}}\{h_\phi(x, \hat{z}) + \gamma(\hat{z})\}$ and $z^*(\epsilon) \triangleq \arg\max_{\hat{z}\in\mathcal{Z}}\{\epsilon f_\theta(x, \hat{z}) + h_\phi(x, \hat{z}) + \gamma(\hat{z})\}$ be two random variables. Then*

$$\nabla_\phi E_\gamma[f_\theta(x, z^*)] = \lim_{\epsilon\to 0} \frac{1}{\epsilon}\Big( E_\gamma[\nabla_\phi h_\phi(x, z^*(\epsilon)) - \nabla_\phi h_\phi(x, z^*)] \Big) \qquad (7)$$

*Proof sketch:* We use a *prediction generating function* $G(\phi, \epsilon) = E_\gamma[\max_{\hat{z}\in\mathcal{Z}}\{\epsilon f_\theta(x, \hat{z}) + h_\phi(x, \hat{z}) + \gamma(\hat{z})\}]$, whose derivatives are functions of the predictions $z^*, z^*(\epsilon)$. The proof is composed of

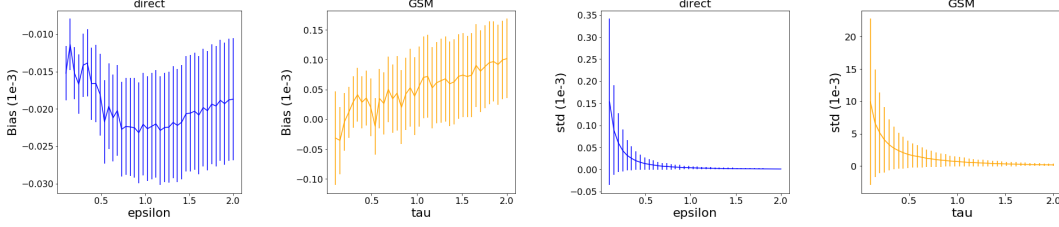

Figure 1: Highlights the the bias-variance tradeoff of the direct optimization estimate as a function of $\epsilon$, compared to the Gumbel-Softmax gradient estimate as a function of its temperature $\tau$. In both cases, the architecture consists of an encoder $X \to FC(300) \to ReLU \to FC(K)$ and a matching decoder. The parameters were learned using the unbiased gradient in Equation (2) to ensure both the direct and GSM have the same (unbiased) reference point. From its optimal parameters we estimate the gradient randomly for $10,000$ times. Left: the bias from the analytic gradient. Right: the average standard deviation of the gradient estimate.

three steps: (i) We prove that $G(\phi, \epsilon)$ is a smooth function of $\phi, \epsilon$. Therefore, the Hessian of $G(\phi, \epsilon)$ exists and it is symmetric, namely $\partial_\phi \partial_\epsilon G(\phi, \epsilon) = \partial_\epsilon \partial_\phi G(\phi, \epsilon)$. (ii) We show that the encoder gradient is apparent in the Hessian: $\partial_\phi \partial_\epsilon G(\phi, 0) = \nabla_\phi E_\gamma [f_\theta(x, z^*)]$. (iii) We rely on the smoothness $G(\phi, \epsilon)$ and derive our update rule as the complement representation of the Hessian: $\partial_\epsilon \partial_\phi G(\phi, 0) = \lim_{\epsilon \to 0} \frac{1}{\epsilon} (E_\gamma [\nabla_\phi h_\phi(x, z^*(\epsilon)) - \nabla_\phi h_\phi(x, z^*)])$. The complete proof is included in the supplementary material. $\qquad\square$

The gradient estimator proposed in Theorem 1 requires two maximization operations. While computing $z^*$ is straightforward, realizing $z^*(\epsilon)$ requires evaluating $f_\theta(x, z)$ for each $z \in \mathcal{Z}$, i.e. evaluating the decoder network multiple times. Nevertheless, the resulting computational overhead can be reduced by performing these operations in parallel (we used batched operations in our implementation).

The gradient estimator is unbiased in the limit $\epsilon \to 0$. However, for small $\epsilon$ values the gradient is either zero, when $z^*(\epsilon) = z^*$, or very large, since the gradients' difference is multiplied by $1/\epsilon$. In practice we use $\epsilon \geq 0.1$ which means that the gradient estimator is biased. In Figure 1 we compare the bias-variance tradeoff of the direct optimization estimator as a function of $\epsilon$, with the Gumbel-Softmax gradient estimator as a function of its temperature $\tau$. Figure 1 shows that while $\epsilon$ and $\tau$ are the sources of bias in these two estimates, they have different impact in each framework.

Algorithm 1 highlights the proposed approach. Each iteration begins with drawing a minibatch $x$ and computing the corresponding latent representations by mapping $x$ to $h_\phi(x, \hat{z})$ and sampling from the resulting posterior distribution $q_\phi(z|x)$ (lines 3-5). The gradients w.r.t. $\theta$ are obtained via standard backpropagation (line 7). The gradients w.r.t. $\phi$ are obtained by reusing the computed $z^*$ (line 5) and evaluating the loss-perturbed predictor (lines 6, 8).

Notably, the $\arg\max$ operations can be solved via non-differentiable solvers (e.g. branch and bound, max-flow).

---

**Algorithm 1** Direct Optimization for discrete VAEs

---

1: $\phi, \theta \leftarrow$ Initialize parameters
2: **while** $\phi, \theta$ not converged **do**
3: $\quad x \leftarrow$ Random minibatch
4: $\quad \gamma \leftarrow$ Random variables drawn from Gumbel distribution.
5: $\quad z^* \leftarrow \arg\max_{\hat{z}} \{h_\phi(x, \hat{z}) + \gamma(\hat{z})\}$
6: $\quad z^*(\epsilon) \leftarrow \arg\max_{\hat{z}} \{\epsilon f_\theta(x, \hat{z}) + h_\phi(x, \hat{z}) + \gamma(\hat{z})\}$
7: $\quad$ Compute $\theta$-gradient:

$$g_\theta \leftarrow \nabla_\theta f_\theta(x, z^*)$$

8: $\quad$ Compute $\phi$-gradient (eq. 7):

$$g_\phi \leftarrow \frac{1}{\epsilon} \left( \nabla_\phi h_\phi(x, z^*(\epsilon)) - \nabla_\phi h_\phi(x, z^*) \right)$$

9: $\quad \phi, \theta \leftarrow$ Update parameters using gradients $g_\phi, g_\theta$
10: **end while**

---

## 4.1 Structured latent spaces

Discrete latent variables often carry semantic meaning. For example, in the CelebA dataset there are $n$ possible attributes for an images, e.g., Eyeglasses, Smiling, see Figure 5. Assigning a binary random variable to each of the attributes, namely $z = (z_1, ..., z_n)$, allows us to generate images with certain attributes turned on or off. In this example, the number of possible realizations of $z$ is $2^n$.

Learning a discrete structured space may be computationally expensive. The Gumbel-Softmax estimator, as described in Equation (4), depends on the softmax normalization constant that requires to sum over exponential many terms (exponential in $n$). This computational complexity can be relaxed by ignoring structural relations within the encoder $h_\phi(x, z)$ and decompose it according to its dimensions, i.e., $h_\phi(x, z) = \sum_{i=1}^n h_i(x, z_i; \phi)$. In this case the normalization constant requires only linearly many term (linear in $n$). However, the encoder does not account for correlations between the variables in the structured latent space.

Gumbel-Max reparameterization can account for structural relations in the latent space $h_\phi(x, z)$ without suffering from the exponential cost of the softmax operation, since computing the $\arg\max$ is often more efficient than summing over all exponential possible options.

For computational efficiency we model only pairwise interactions in the structured encoder:

$$h_\phi(x, z) = \sum_{i=1}^n h_i(x, z_i; \phi) + \sum_{i,j=1}^n h_{i,j}(x, z_i, z_j; \phi) \tag{8}$$

The additional modeling power of $h_{i,j}(x, z_i, z_j; \phi)$ allows the encoder to better calibrate the dependences of the structured latent space that are fed into the decoder. In general, the pairwise correlations requires a quadratic integer program solvers, such as the CPLEX to recover the $\arg\max$. However, efficient maxflow solvers may be used when the pairwise correlations have special structural restrictions, e.g., $h_{i,j}(x, z_i, z_j; \phi) = \alpha_{i,j}(x) z_i z_j$ for $\alpha_{i,j}(x) \geq 0$.

The gradient realization in Theorem 1 holds also for the structured setting, whenever the structure of $\gamma$ follows the structure of $h_\phi$. This gradient realization requires to compute $z^*$, $z^*(\epsilon)$. While $z^*$ only depends on the structured encoder, the $\arg\max$-perturbation $z^*(\epsilon)$ involves the structured decoder $f_\theta(x, z_1, ..., z_n)$ that does not necessarily decompose according to the structured encoder. We use the fact that we can compute $z^*$ efficiently and apply the low dimensional approximation $\tilde{f}_\theta(x, z) = \sum_{i=1}^n \tilde{f}_i(x, z_i; \theta)$, where $\tilde{f}_i(x, z_i; \theta) = f_\theta(x, z_1^*, ..., z_i, ..., z_n^*)$. With this in mind, we approximate $z^*(\epsilon)$ with $\tilde{z}^*(\epsilon)$ that is computed by replacing $f_\theta(x, z)$ with $\tilde{f}_\theta(x, z)$. In our implementation we use the batch operation to compute $\tilde{f}_\theta(x, z)$ efficiently.

## 4.2 Semi-supervised learning

Direct optimization naturally extends to semi-supervised learning, where we may add to the learning objective the loss function $\ell(z, z^*)$, for supervised samples $(x, z) \in S_1$, to better control the prediction of the latent space. The semi-supervised discrete VAEs objective function is

$$\sum_{x \in S} \mathbb{E}_\gamma[f_\theta(x, z^*)] + \sum_{(x,z) \in S_1} \mathbb{E}_\gamma[\ell(z, z^*)] + \sum_{x \in S} KL(q_\phi(z|x)||p_\theta(z)) \tag{9}$$

The supervised component is explicitly handled by Theorem 1. Our supervised component is intimately related to direct loss minimization [24, 35]. The added random perturbation $\gamma$ allows us to use a generative model to prediction, namely, we can randomly generate different explanations $z^*$ while the direct loss minimization allows a single explanation for a given $x$.

# 5 Experimental evaluation

We begin our experiments by comparing the test loss of direct optimization, the Gumbel-Softmax (GSM) and the unbiased gradient computation in Equation (2). We performed these experiments using the binarized MNIST dataset [33], Fashion-MNIST [40] and Omniglot [20]. The architecture consists of an encoder $X \rightarrow FC(300) \rightarrow ReLU \rightarrow FC(K)$, a matching decoder $K \rightarrow FC(300) \rightarrow ReLU \rightarrow FC(X)$ and a BCE loss. Following [12] we set our learning rate to $1e-3$ and the annealing rate to $1e-5$ and we used their annealing schedule every 1000 steps, setting the minimal $\epsilon$ to be 0.1. The results appear in Table 1. When considering MNIST and Omniglot, direct optimization achieves similar test loss to the unbiased method, which uses the analytical gradient computation in Equation (2). Also, direct optimization achieves a better result than GSM, in spite the fact both direct optimization and GSM use biased gradient descent: direct optimization uses a biased gradient for the exact objective in Equation (1), while GSM uses an exact gradient for an approximated objective. Surprisingly, on Fashion-MNIST, direct optimization achieves better test loss than the unbiased. To

| $k$ | MNIST | | | Fashion MNIST | | | Omniglot | | |
|---|---|---|---|---|---|---|---|---|---|
| | unbiased | direct | GSM | unbiased | direct | GSM | unbiased | direct | GSM |
| 10 | 164.53 | 165.26 | 167.88 | 228.46 | 222.86 | 238.37 | 155.44 | 155.94 | 160.13 |
| 20 | 152.31 | 153.08 | 156.41 | 206.40 | 198.39 | 211.87 | 152.05 | 152.13 | 166.76 |
| 30 | 149.17 | 147.38 | 152.15 | 205.60 | 189.44 | 197.01 | 152.10 | 150.14 | 157.33 |
| 40 | 142.86 | 143.95 | 147.56 | 205.68 | 184.21 | 195.22 | 151.38 | 150.33 | 156.09 |
| 50 | 155.37 | 140.38 | 146.12 | 200.88 | 180.31 | 191.00 | 156.84 | 149.12 | 164.01 |

Table 1: Compares the test loss of VAEs with different categorial variables $z \in \{1, ..., k\}$. Direct optimization achieves similar test loss to the unbiased method (Equation (2)) and achieves a better test loss than GSM, in spite the fact both direct optimization and GSM use biased gradient descent.

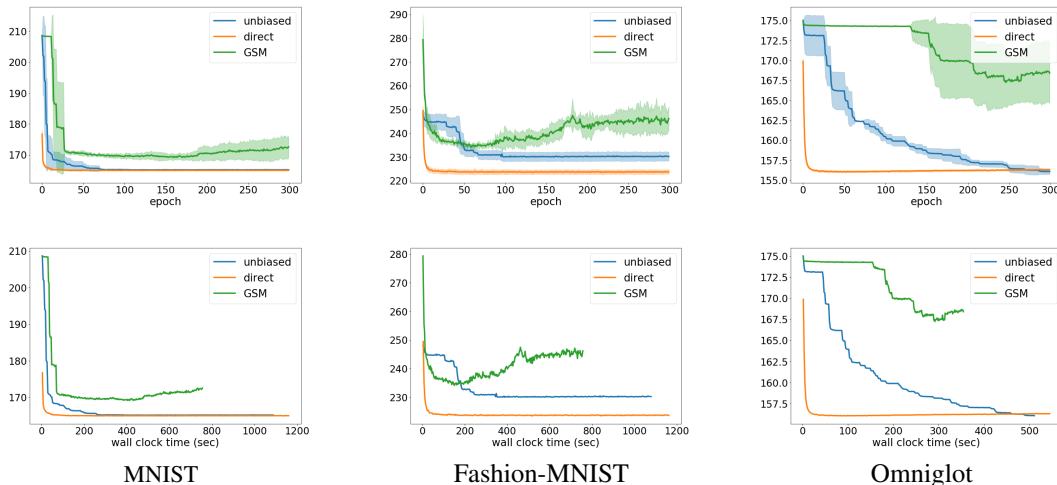

MNIST            Fashion-MNIST            Omniglot

Figure 2: Comparing the decrease of the test loss for $k = 10$. Top row: test loss as a function of the learning epoch. Bottom row: test loss as a function of the learning wall-clock time. Incomplete plot in the bottom row suggests the algorithm required less time to finish 300 epochs.

further explore this phenomenon, in Figure 2 one can see that the unbiased method takes more epochs to converge, and eventually it achieves similar and often better test loss than direct optimization on MNIST and Omniglot. In contrast, on Fashion-MNIST, direct optimization is better than the unbiased gradient method, which we attribute to the slower convergence of the unbiased method, see supplementary material for more evidence.

It is important to compare the wall-clock time of each approach. The unbiased method requires $k$ computations of the encoder and the decoder in a forward and backward pass. GSM requires a single forward pass and a single backward pass (encapsulating the $k$ computations of the softmax normalization within the code). In contrast, our approach requires a single forward pass, but $k$ computations of the decoder $f_\theta(x, z)$ for $z = 1, ..., k$ in the backward pass. In our implementation we use the batch operation to compute $f_\theta(x, z)$ efficiently. Figure 2 compares the test loss as a function of the wall clock time and shows that while our method is 1.5 times slower than GSM, its test loss is lower than the GSM at any time.

Next we perform a set of experiments on Fashion-MNIST using discrete structured latent spaces $z = (z_1, ..., z_n)$ while each $z_i$ is binary, i.e., $z_i \in \{0, 1\}$. In the following experiments we consider a structured decoder $f_\theta(x, z) = f_\theta(x, z_1, ..., z_n)$. The decoder architecture consists of the modules $(2 \times 15) \to FC(300) \to ReLU \to FC(X)$ and a BCE loss. For $n = 15$ the computational cost of the softmax in GSM is high (exponential in $n$) and therefore one cannot use a structured encoder with GSM.

Our first experiment with a structured decoder considers an unstructured encoder $h_\phi(x, z) = \sum_{i=1}^{n} h_i(x, z_i; \phi)$ for GSM and direct optimization. This experiment demonstrates the effectiveness of our low dimensional approximation $\tilde{f}_\theta(x, z) = \sum_{i=1}^{n} \tilde{f}_i(x, z_i; \theta)$, where $\tilde{f}_i(x, z_i; \theta) =$

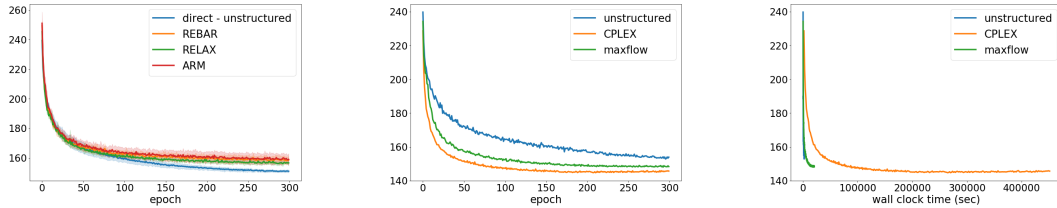

Figure 3: Left: test loss of unstructured encoder and a structured decoder as a function of their epochs. Middle: using structured decoders and comparing unstructured encoders to structured encoders, $h_{i,j}(x, z_i, z_j; \phi) = \alpha_{i,j}(x)z_i z_j$, both for general $\alpha_{i,j}(x)$ (recovering the $\arg\max$ using CPLEX) and for $\alpha_{i,j}(x) \geq 0$ (recovering the $\arg\max$ using maxflow). Right: comparing the wall-clock time of decomposable and structured encoders.

| #labels | MNIST | | | | Fashion-MNIST | | | |
| | accuracy | | bound | | accuracy | | bound | |
| | direct | GSM | direct | GSM | direct | GSM | direct | GSM |
|---|---|---|---|---|---|---|---|---|
| 50 | 92.6% | 84.7% | 90.24 | 91.23 | 63.3% | 61.2% | 129.66 | 129.813 |
| 100 | 95.4% | 88.4% | 90.93 | 90.64 | 67.2% | 64.2% | 130.822 | 129.054 |
| 300 | 96.4% | 91.7% | 90.39 | 90.01 | 70.0% | 69.3% | 130.653 | 130.371 |
| 600 | 96.7% | 92.3% | 90.78 | 89.77 | 72.1% | 71.6% | 130.81 | 129.973 |
| 1200 | 96.8% | 92.7% | 90.45 | 90.37 | 73.7% | 73.2% | 130.921 | 130.063 |

Table 2: Semi-supervised VAE on MNIST and Fashion-MNIST with $50/100/300/600/1200$ labeled examples out of the $50,000$ training examples.

$f_\theta(x, z_1^*, ..., z_i, ..., z_n^*)$ for applying direct optimization to structured decoders in Section 4.1. We also compare the unbiased estimators REBAR [36] and RELAX [9] and the recent ARM estimator [41].[3] The results appear in Figure 3 and may suggest that using the approximated $\tilde{z}^*(\epsilon)$, the gradient estimate of direct optimization still points towards a direction of descent for the exact objective.

Our second experiment uses a structured decoder with structured encoders, which may account for correlations between latent random variables $h_\phi(x, z) = \sum_{i=1}^{n} h_i(x, z_i; \phi) + \sum_{i,j=1}^{n} h_{i,j}(x, z_i, z_j; \phi)$. In this experiment we compare two structured encoders with pairwise functions $h_{i,j}(x, z_i, z_j; \phi) = \alpha_{i,j}(x)z_i z_j$. We use a general pairwise structured encoder where the $\arg\max$ is recovered using the CPLEX algorithm [6]. We also apply a super-modular encoder, where $\alpha_{i,j}(x) \geq 0$ is enforced using the softplus transfer function, and the $\arg\max$ is recovered using the maxflow algorithm [4]. In Figure 3 we compare the general and super-modular structured encoders with an unstructured encoder ($\alpha_{i,j}(x) = 0$), all are learned using direct optimization. One can see that structured encoders achieve better bounds, while the wall-clock time of learning super-modular structured encoder using maxflow ($\alpha_{i,j}(x) \geq 0$) is comparable to learning unstructured encoders. One can also see that the general structured encoder, with any $\alpha_{i,j}(x)$, achieves better test loss than the super-modular structured encoder. However, this comes with a computational price, as the maxflow algorithm is orders of magnitude faster than CPLEX, and structured encoder with CPLEX becomes better than maxflow only in epoch 85, see Figure 3.

Finally, we perform a set of semi-supervised experiments, for which we use a mixed continuous discrete architecture, [15, 12]. The architecture of the base encoder is $(28 \times 28) \rightarrow FC(400) \rightarrow ReLU \rightarrow FC(200)$. The output of this layer is fed both to a discrete encoder $h_d$ and a continuous encoder $h_c$. The discrete latent space is $z_d \in \{1, ..., 10\}$ and its encoder $h_d$ is $200 \rightarrow FC(100) \rightarrow ReLU \rightarrow FC(10)$. The continuous latent space considers $k = 10, c = 20$, and its encoder $h_c$ consists of a $200 \rightarrow FC(100) \rightarrow ReLU \rightarrow FC(66) \rightarrow FC(40)$ to estimate the mean and variance of $20$−dimensional Gaussian random variables $z_1, ..., z_{10}$. The mixed discrete-continuous latent space consists of the matrix $diag(z_d^*) \cdot z_c$, i.e, if $z_d^* = i$ then this matrix is all zero, except for the $i$-th row. The parameters of $z_c$ are shared across the rows $z = 1, ..., k$ through the batch operation.

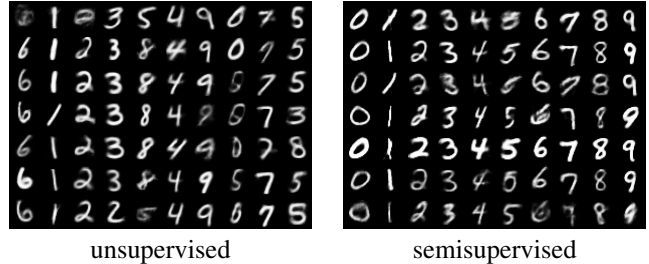

|           unsupervised           |           semisupervised           |

Figure 4: Comparing unsupervised to semi-supervised VAE on MNIST, for which the discrete latent variable has 10 values, i.e., $z \in \{1, ..., 10\}$. Weak supervision helps the VAE to capture the class information and consequently improve the image generation process.

| w/o glasses | | | | glasses | | | |
| woman | | man | | woman | | man | |
| w/o smile | smile | w/o smile | smile | w/o smile | smile | w/o smile | smile |
| 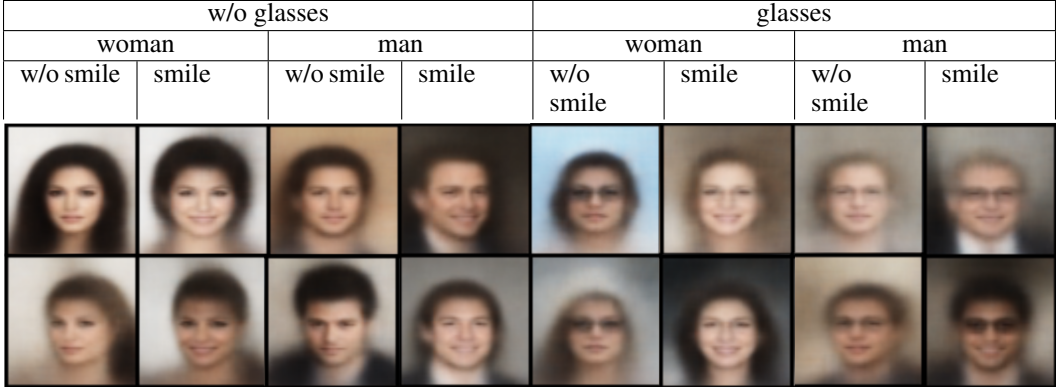 | | | | | | | |

Figure 5: Learning attribute representation in CelebA, using our semi-supervised setting, by calibrating our $\arg\max$ prediction using a loss function. These images here are generated while setting their attributes to get the desired image. The $i-$th row consists the generation of the same continuous latent variable for all the attributes

We conducted a quantitive experiment with weak supervision on MNIST and Fashion-MNIST with $50/100/300/600/1200$ labeled examples out of the $50,000$ training examples. For labeled examples, we set the perturbed label $z^*(\epsilon)$ to be the true label. This is equivalent to using the indicator loss function over the space of correct predictions. A comparison of direct optimization with GSM appears in Table 2. Figure 4 shows the importance of weak supervision in semantic latent space, as it allows the VAE to better capture the class information.

Supervision in generative models also helps to control discrete semantics within images. We learn to generate images using $k = 8$ discrete attributes of the CelebA dataset (cf. [22]) while using our semi-supervised VAE. For this task, we use convolutional layers for both the encoder and the decoder, except the last two layers of the continuous latent model which are linear layers that share parameters over the 8 possible representations of the image. In Figure 5, we show generated images with discrete semantics turned on/off (with/without glasses, with/without smile, woman/man).

## 6 Discussion and future work

In this work, we use the Gumbel-Max trick to reparameterize discrete VAEs using the $\arg\max$ prediction and show how to propagate gradients through the non-differentiable $\arg\max$ function. We show that this approach compares favorably to state-of-the-art methods, and extend it to structured encoders and semi-supervised learning.

These results can be taken in a number of different directions. Our gradient estimation is practically biased, while REINFORCE is an unbiased estimator. As a result, our methods may benefit from the REBAR/RELAX framework, which directs biased gradients towards the unbiased gradient [36, 31]. There are also optimization-related questions that arise from our work, such as exploring the interplay between the $\epsilon$ parameter and the learning rate.

## Footnotes

[1]The set $\arg\max_{\hat{z} \in \mathcal{Z}}\{h_\phi(x, \hat{z}) + \gamma(\hat{z})\}$ contains all maximizing assignments (possibly more than one). However, since the Gumbel distribution is continuous, the $\gamma$ for which the set of maximizing assignments contains multiple elements has measure zero. For notational convenience, when we consider integrals (or probability distributions), we ignore measure zero sets.

[2]We match the notation of the parameters $\phi$ of the posterior distribution to highlight the connection between the two objectives.

[3]For REBAR and RELAX we used the code in `https://github.com/duvenaud/relax`. and for ARM we used the code in `https://github.com/mingzhang-yin/ARM-gradient`

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
