[Supplementary Material · supplementary material.pdf]

# Supplementary material: Direct Optimization through $\arg\max$ for Discrete Variational Auto-Encoder

**Guy Lorberbom**
Technion

**Andreea Gane**
MIT

**Tommi Jaakkola**
MIT

**Tamir Hazan**
Technion

**Theorem 1.** *Assume $h_\phi(x, z)$ is a smooth function of $\phi$. Let $z^* \triangleq \arg\max_{\hat{z}}\{h_\phi(x, \hat{z}) + \gamma(\hat{z})\}$ and $z^*(\epsilon) \triangleq \arg\max_{\hat{z}}\{\epsilon f_\theta(x, \hat{z}) + h_\phi(x, \hat{z}) + \gamma(\hat{z})\}$ be two random variables. Then*

$$\nabla_\phi E_\gamma[f_\theta(x, z^*)] = \lim_{\epsilon \to 0} \frac{1}{\epsilon}\Big( E_\gamma[\nabla_\phi h_\phi(x, z^*(\epsilon)) - \nabla_\phi h_\phi(x, z^*)]\Big) \tag{1}$$

*Proof.* We use a "prediction generating function" $G(\phi, \epsilon) = E_\gamma[\max_{\hat{z}}\{\epsilon f_\theta(x, \hat{z}) + h_\phi(x, \hat{z}) + \gamma(\hat{z})\}]$, whose derivatives are functions of the predictions $z^*, z^*(\epsilon)$. The proof is composed from three steps:

1. We prove that $G(\phi, \epsilon)$ is a smooth function of $\phi, \epsilon$. Therefore, the Hessian of $G(\phi, \epsilon)$ exists and it is symmetric, namely

$$\partial_\phi \partial_\epsilon G(\phi, \epsilon) = \partial_\epsilon \partial_\phi G(\phi, \epsilon). \tag{2}$$

2. We show that encoder gradient is apparent in the Hessian:

$$\partial_\phi \partial_\epsilon G(\phi, 0) = \nabla_\phi E_\gamma[\theta(x, z^*)]. \tag{3}$$

3. We derive our update rule as the complement representation of the Hessian:

$$\partial_\epsilon \partial_\phi G(\phi, 0) = \lim_{\epsilon \to 0} \frac{1}{\epsilon}\Big( E_\gamma[\nabla_\phi h(x, z^*(\epsilon)) - \nabla_\phi h(x, z^*)]\Big) \tag{4}$$

First, we prove that $G(\phi, \epsilon)$ is a smooth function. Recall, $g(\gamma) = \prod_{z=1}^{k} e^{-(\gamma(z)+c+e^{-(\gamma(z)+c)})}$ is the zero mean Gumbel probability density function. Applying a change of variable $\hat{\gamma}(z) = \epsilon f_\theta(x, \hat{z}) + h_\phi(x, \hat{z}) + \gamma(\hat{z})$, we obtain

$$G(\phi, \epsilon) = \int_{\mathbb{R}^k} g(\gamma)\max_{\hat{z}}\{\epsilon f_\theta(x, \hat{z}) + h_\phi(x, \hat{z}) + \gamma(\hat{z})\}d\gamma = \int_{\mathbb{R}^k} g(\hat{\gamma} - \epsilon f_\theta - h_\phi)\max_{\hat{z}}\{\hat{\gamma}(\hat{z})\}d\hat{\gamma}.$$

Since $g(\hat{\gamma} - \epsilon f_\theta - h_\phi)$ is a smooth function of $\epsilon$ and $h_\phi(x, z)$ and $h_\phi(x, z)$ is a smooth function of $\phi$, we conclude that $G(\phi, \epsilon)$ is a smooth function of $\phi, \epsilon$. Therefore, the Hessian of $G(\phi, \epsilon)$ exists and symmetric, i.e., $\partial_\phi \partial_\epsilon G(\phi, \epsilon) = \partial_\epsilon \partial_\phi G(\phi, \epsilon)$. We thus proved Equation (2).

To prove Equations (3) and (4) we differentiate under the integral, both with respect to $\epsilon$ and with respect to $\phi$. We are able to differentiate under the integral, since $g(\hat{\gamma} - \epsilon f_\theta - h_\phi)$ is a smooth function of $\epsilon$ and $\phi$ and its gradient is bounded by an integrable function (cf. [2], Theorem 2.27, using the continuity of the max function).

We turn to prove Equation (3). We begin by noting that $\max_{\hat{z}}\{\epsilon f_\theta(x, \hat{z}) + h_\phi(x, \hat{z}) + \gamma(\hat{z})\}$ is a maximum over linear function of $\epsilon$, thus by Danskin Theorem (cf. [1], Proposition 4.5.1) holds $\partial_\epsilon(\max_{\hat{z}}\{\epsilon f_\theta(x, \hat{z}) + h_\phi(x, \hat{z}) + \gamma(\hat{z})\}) = f_\theta(x, z^*(\epsilon))$. By differentiating under the integral, $\partial_\epsilon G(\phi, \epsilon) = \mathbb{E}_\gamma[f_\theta(x, z^*(\epsilon))]$. We obtain Equation (3) by differentiating under the integral, now with respect to $\phi$, and setting $\epsilon = 0$.

Figure 1: Test loss for $k = 20, 30, 40, 50$ (left: MNIST, middle: Fashion-MNIST, right: Omniglot)

Finally, we turn to prove Equation (4). By differentiating under the integral $\partial_\phi G(\phi, \epsilon) = \mathbb{E}_\gamma[\nabla_\phi h_\phi(x, z^*(\epsilon))]$. Equation (4) is attained by taking the derivative with respect to $\epsilon = 0$ on both sides.

The theorem follows by combining Equation (2) when $\epsilon = 0$, i.e., $\partial_\phi \partial_\epsilon G(\phi, 0) = \partial_\epsilon \partial_\phi G(\phi, 0)$ with the equalities in Equations (3) and (4). $\qquad \square$

# 1    Gumbel-Max perturbation model and the Gibbs distribution

**Theorem 2.** *[3, 4, 5] Let $\gamma$ be a random function that associates random variable $\gamma(z)$ for each $z = 1, ..., k$ whose distribution follows the zero mean Gumbel distribution law, i.e., its probability density function is $g(t) = e^{-(t+c+e^{-(t+c)})}$ for the Euler constant $c \approx 0.57$. Then*

$$\frac{e^{h_\phi(x,z)}}{\sum_{\hat{z}} e^{h_\phi(x,\hat{z})}} = \mathbb{P}_{\gamma \sim g}[z = z^*],$$

$$where \; z^* \triangleq \arg \max_{\hat{z}=1,...,k} \{h_\phi(x, \hat{z}) + \gamma(\hat{z})\} \tag{5}$$

*Proof.* Let $G(t) = e^{-e^{-(t+c)}}$ be the Gumbel cumulative distribution function. Then

$$
\begin{aligned}
\mathbb{P}_{\gamma \sim g}[z = z^*] &= \mathbb{P}_{\gamma \sim g}[z = \arg \max_{\hat{z}=1,..,k} \{h_\phi(x, \hat{z}) + \gamma(\hat{z})\}] \\
&= \int g(t - \phi(x, z)) \prod_{\hat{z} \neq z} G(t - h_\phi(x, \hat{z})) dt
\end{aligned}
$$

Since $g(t) = e^{-(t+c)}G(t)$ it holds that

$$\int g(t - h_\phi(z)) \prod_{\hat{z} \neq z} G(t - h_\phi(\hat{z}))dt \tag{6}$$

$$= \int e^{-(t - h_\phi(x,z) + c)} G(t - h_\phi(x,z)) \prod_{\hat{z} \neq z} G(t - h_\phi(x, \hat{z}))dt$$

$$= \frac{e^{h_\phi(x,z)}}{Z} \tag{7}$$

where $\frac{1}{Z} = \int e^{-(t+c)} \prod_{\hat{z}=1}^{k} G(t - h_\phi(\hat{z}))dt$ is independent of $z$. Since $\mathbb{P}_{\gamma \sim g}[z = z^*]$ is a distribution then $Z$ must equal to $\sum_{\hat{z}=1}^{k} e^{h_\phi(x, \hat{z})}$. $\qquad\square$