[Reviews · NeurIPS 2019]

Reviewer 1



The authors present a novel approach for optimizing discrete latent variable models. The approach is a straight-forward combination of the recently introduced direct loss minimization technique (originally designed for structured prediction) with Gumbel-max re-parameterization. This approach avoids the approximation to the arg-max that other methods employ, making in conceptually attractive. The authors apply the method to several datasets and a few uses cases (semi-supervised learning, learning structured latent distributions) and show competitive performance compared to existing methods. Optimizing discrete latent variable models is a basic problem with broad applicability. Methods for optimizing such models is an area of active research. Given the original approach, this work is likely to be of interest to many in the NeurIPS community. The submission appears technically sounds in all of its derivations and the results show competitive or superior results when compared to existing methods. The clarity of the work is decent but certainly could be improved. Since the approach is a relatively straightforward application of two existing methods (Gumbel-max/ direct loss minimization) a more thorough and clear summary of those methods as a background would increase the clarity of the submission and allow a more general audience to fully appreciate and understand the work. Presently, serious consultations with referenced methods are required, even for a reader fairly acquainted with the general approach. The originality of the approach, its technical quality and significance make the paper worthy of acceptance, but the clarity of the presentation detracts from it. Given that the approach appears to be the straightforward application and combination of two existing ideas, the onus falls on the authors to distill and explain these ideas to the reader.

Reviewer 2



Originality =========== To the best of my knowledge, the insight that the Gumbel-max trick can be combined with direct loss mimization in the VAE setting is novel. Moreover, the work that had to be done to get the ideas to play well together seems significant and original. Quality =========== I have some reservations about the method presented, which is why I've given a slightly negative overall score. However, it seems very plausible that an author response could clarify my concerns and cause me to revise my score upward. Figure 1: -Epsilon and Tau are different variables in quite different settings. It seems weird to have them share an axis. I'd find the chart more intuitive if one chart showed the bias and variance for GSM and one for Direct. -I would appreciate some error bars for standard error of your bias and variance estimates. ----The authors have satisfied me entirely on this suggestion. Though I would suggest rewording some of the text based on the very wide error bars on Figure 1.---- Theorem 1: -I haven't compared to the direct loss paper [28] in immense detail, but I notice that in their proof there are no gradients inside the expectation on the RHS. Do you have an intuitive explanation for why you have gradients here and they don't? You imply a correspondance in "The above therem closely relates to ..." but as far as I can see plugging this loss into their paper would not give your estimator. ----The authors have corrected my misunderstanding of this point. Thank you.---- -I'm slightly puzzled by the final step of your proof. You set two terms equal to each other by setting epsilon=0, but in the same breath you take the limit of epsilon to 0. I'm not sure you can do both at the same time. ----I think I'm happy with this on reflection---- -I would appreciate more discussion of the nature of the bias that you introduce by using epsilon != 0. ----I still think that developing this further would improve the paper, but it is good enough without it.---- Experiments comparing loss, e.g., Table 1 and Figure 2: -I'm very puzzled by the fact that your unbiased estimator is beaten by direct and even GSM as k goes up. You say that this is due to the relative complexity of FashionMNIST, but the effect exists in all the other datasets at k=50 and Fashion MNIST isn't more complex than Omniglot. You say it may be due to slower convergence, but did the effect go away when you ran it longer? The models look fairly converged in your plot. You say it may be due to the non-convexity of the bound, but why would this affect direct less than unbiased? Can you explain this remark a little more? My concern is that something odd may be going on that casts the experimental evidence into a little more doubt. -Also puzzled that the loss for direct and GSM are higher for k=50 than k=40 for MNIST and Omniglot. Do you have an explanation for that? ----I'm still a bit unsure about what's going on here, but the results seem important and sufficiently solid despite that.---- -Nice-to-have: I'd be interested to see the effect of changing epsilon on your results. ----This would still be nice, but is fine.---- -I'd appreciate standard error and averaging over multiple random seeds. ----Thank you for including this.---- The experiments with semi-supervised loss or structured encoders seem good. I would love to have some comparisons for Figure 5 and 4 to your baselines, be it GSM or unbiased. Not including them makes me wonder if there isn't a detectable improvement, which makes me wonder if the result is sufficiently significant. ----I understand that these are qualitative and that there are comparisons elsewhere. I was and am still curious about what the practical significance of the new method was relative to baselines on these results. What is already here is obviously fine as far as it goes, but comparing to baselines could make the point stronger.---- Clarity =========== I think you could cut almost all of the 2nd para on p1 since it repeats itself in the related work section. I found your notation for z slightly confusing. Sometimes you use it as a random variable, and sometimes you use it as an index/the values the random variable might take on. This results also in your notation in S3 being inconsistent with your later work where z is a set of binary random variables. I think you could improve the legibility of your paper a bit by being clearer about these things. Most graphs have far too small text. Small typos: -p1 "applied to structure setting[s]" -p4 "since the gradients" probably should have an apostrophe -In the future could you please use the submission option of the Neurips package. This gives reviewers line-numbers which make it easier to give feedback. Significance =========== Low variance estimators for gradients with discrete latent variables of significant interest to the community. I am satisfied by the experimental work. There are some slightly unexpected behaviours with the unbiased baseline, but I'm satisfied that they don't reflect a hidden weakness of the presented approach.

Reviewer 3



Reparameterization of VAE with continuous random variables has been developed well, no matter from theory or application views. In this paper, the authors provide an optimization technique that propagates (biased) gradients through the reparameterized argmax. Based on it, the authors develop discrete VAE. The manuscript is full of theoretical analysis and experimental results to help readers understand their method and motivation. I have some small suggestions. 1) The size of legend in third subfigure in Fig.3 should be consistent with that in other subfigures. 2) In figure 5, the authors choose 8 discrete attributes of CelebA to learn the model. They only exhibit three simple semantics turned on/off. Could you give more results in the Supplementary material? 3) The providing code is good. Could you provide an algorithm of the method in the paper to help the readers, who do not have time to read the code, understand the work?

[Author Response · NeurIPS 2019]



Figure 1: Bias-Variance tradeoff

Figure 2: Test loss for $K = 50$ (right: MNIST, middle: Fashion-MNIST, left: Omniglot

We thank the reviewers for their time and appreciate the feedback. Are happy all reviewers agree the work is significant and novel ("Novel approach... High significance. General problem, area of active research"; "seems significant and original"; "This contribution is important").

**[R1]** We thank you for your notes and we will use the extra page to provide a more thorough and clear summary of our components, especially direct loss minimization. We will also make a clear distinction between the model parameters and the distributions and we will consolidate our terms about Gumbel-Max trick and Gumbel-Max perturbation models, thank you for emphasizing it in your review.

**[R3]** We thank you for your suggestions and we will revise the work accordingly as well as include an algorithm of the method in the paper. We will also add more results for the CelebA in the supplementary material — we did not include them in this response due to lack of space.

**[R2]** $\epsilon$ **and** $\tau$ **are different variables in quite different settings:** We agree they differ but since their order of magnitude is different, we lose information when placing them in the same plot. In Figure 1 we separated all four options and added error bars. **In the direct loss paper, there are no gradients inside the expectation:** Their work considers linear models $\phi(x, y; w) \stackrel{def}{=} \langle w, \hat{\phi}(x, y) \rangle$ (note, $\phi(x, y; w)$ in our notation is $s_w(x, y)$ in their notation in their paragraph 2) while our work considers non-linear models. Therefore, in their setting $\nabla_w \phi(x, y; w) = \hat{\phi}(x, y)$ and in our setting we consider the general gradient notation $\nabla_w \phi(x, y; w)$. **Puzzled by the final step of your proof:** Following the complete proof in the appendix: $\partial_w G(w, \epsilon) = \mathbb{E}_\gamma [\nabla_w \phi(x, z^{\epsilon\theta + \phi + \gamma}; w)]$. Since we show that $G(w, \epsilon)$ is smooth, then $\partial_w G(w, \epsilon)$ is smooth and its derivative with respect to $\epsilon$ existsThis derivative is defined by the limit operation. **Puzzled by the fact that your unbiased estimator is beaten by direct and even GSM as k goes up:** We agree that this result is surprising and we also provide code so this behavior can be verified. In Figure 2 in this response we present plots for $k = 50$ to complement the results in the paper ($k = 10$). We see that: (i) the gap is the largest for Fashion-MNIST. (ii) when we let the unbiased algorithm to run for long enough the gap between the unbiased minimization and the GSM and direct is getting smaller (iii) the unbiased algorithm sometimes reduces the test loss in steps, so it is quite possible that if we let the unbiased to run for longer we might see eventually more reductions in the test loss till its performance is the same as direct and GSM. (iv) It is certainly possible that this gap will remain even if we run the algorithms for longer time, as these algorithms optimize a non-convex function. We conjecture that if this is the case then it might be on par with the difference between gradient descent (unbiased) and stochastic gradient descent (direct, GSM) that more strongly pursue direction of decent. We will include all plots ($k = 10, 20, .., 50$) to make this behavior apparent. **Loss for direct and GSM are higher for k=50 than k=40 for Omniglot:** Perhaps with the added parameters a worse minimum was found. We kindly point out that the direct and GSM still improve in $k = 50$ over $k = 40$ in MNIST and Fashion-MNIST. **I would love to have some comparisons for Figure 5 and 4 to your baselines:** These figures are qualitative and generally suggest that supervision helps and provide motivational example for the structured setting. Quantitive results appears in Table 2 and Figure 3. **I'd appreciate standard error and averaging over multiple random seeds:** We ran this experiment and omitted due to lack of space. We will add it to the supplementary material. We will follow the suggestions about the clarity.

[Meta-Review · NeurIPS 2019]

The reviewers arrived at a consensus and recommend to accept this submission. Following the reviewers' request, please improve clarity for the camera ready version of the paper.